# Studying Chondrichthyans Using Baited Remote Underwater Video Systems: A Review

**DOI:** 10.3390/ani14131875

**Published:** 2024-06-25

**Authors:** Francesco Luigi Leonetti, Massimiliano Bottaro, Gianni Giglio, Emilio Sperone

**Affiliations:** 1Department of Biology, Ecology and Earth Sciences, University of Calabria, 87036 Rende, Italy; gianni.giglio@unical.it; 2Genoa Marine Centre, Department of Integrative Marine Ecology (EMI), Stazione Zoologica Anton Dohrn, Italian National Institute for Marine Biology, Ecology and Biotechnology, Villa del Principe, Piazza del Principe 4, 16126 Genoa, Italy; massimiliano.bottaro@szn.it

**Keywords:** BRUVS, cartilaginous fish, ecology, life history, underwater surveys, conservation

## Abstract

**Simple Summary:**

The study explores the use of Baited Remote Underwater Video Systems (BRUVS) as a non-invasive alternative to traditional capture-based surveys for studying cartilaginous fish populations. Over the past three decades, BRUVS have gained significant popularity, particularly in regions like Australia, South Africa, and Central America. By analyzing 81 peer-reviewed papers from 1990 to 2023, it was found that BRUVS have been crucial in studying 195 species of sharks, rays, and chimeras. These systems allow for standardized surveys across various habitats, minimizing disturbance to marine life. Common BRUVS setups include benthic configurations and mono-camera systems, often using bait from specific fish families. BRUVS offer versatility, allowing for tailored setups to meet different research objectives, such as accurate size estimation and individual identification. Importantly, BRUVS facilitate the monitoring of endangered and data-deficient species, contributing essential data for conservation efforts. Overall, the study highlights BRUVS as a valuable tool for studying and conserving cartilaginous fish populations globally.

**Abstract:**

Cartilaginous fish face significant threats due to overfishing and slow reproductive rates, leading to rapid declines in their populations globally. Traditional capture-based surveys, while valuable for gathering ecological information, pose risks to the health and survival of these species. Baited Remote Underwater Video Systems (BRUVS) offer a non-invasive alternative, allowing for standardized surveys across various habitats with minimal disturbance to marine life. This study presents a comprehensive review of BRUVS applications in studying cartilaginous fish, examining 81 peer-reviewed papers spanning from 1990 to 2023. The analysis reveals a significant increase in BRUVS usage over the past three decades, particularly in Australia, South Africa, and Central America. The most common BRUVS configurations include benthic setups, mono-camera systems, and the use of fish from the Clupeidae and Scombridae families as bait. BRUVS have been instrumental in studying 195 chondrichthyan species, providing insights into up to thirteen different aspects of the life histories. Moreover, BRUVS facilitate the monitoring of endangered and data-deficient species, contributing crucial data for conservation efforts. Overall, this study underscores the value of BRUVS as a powerful tool for studying and conserving cartilaginous fish populations worldwide.

## 1. Introduction

Cartilaginous fish worldwide are in rapid decline due to chronic overfishing and their slow reproductive life-history characteristics [1,2,3,4]. For example, at least 53% of the sharks, rays, and chimeras native to the Mediterranean Sea are at risk of extinction and require urgent action to conserve their populations [5,6]. In this scenario, ecological information such as diversity, distribution, and abundance of Chondrichthyans is crucial for the development of effective management and conservation initiatives [7,8]. However, for most cartilaginous fish, even the most basic ecological information is lacking, which has led to 16% of 1240 species assessed being listed as ‘Data Deficient’ on the IUCN Red List of Threatened Species [9].

The most common method to collect basic ecological information on sharks, rays, and chimeras is through scientific trawl and longline surveys [10]. These surveys generate relative abundance estimates, providing comparative data on abundance through space and time on both inter- and intra-species levels [11]. Despite their utility and widespread application, these surveys have significant drawbacks: they require the capture, retrieval, and restraint of the animals, causing various degrees of physical trauma and physiological stress. These homeostatic disruptions can potentially impact growth, feeding, and the immune system, increasing the chance of mortality [12]. Considering the rapid declines in many Chondrichthyan populations, the consequences of this type of surveys are in contrast with the objectives of many conservation programs.

Baited Remote Underwater Video Systems (BRUVS) are an effective tool for collecting data on Chondrichthyans regarding their occurrence, distribution, abundance, behavior, and other key aspects in outlining their life histories [13]. Data collection involves using high-definition underwater cameras mounted on a frame, with bait, typically commonly fished marine species (fish parts, entrails), and/or other attractants placed in a bait canister to lure Chondrichthyans into the camera’s field of view [11]. BRUVS are deployed at various locations and depths to cover different habitats and environmental conditions. Multiple deployments are conducted at each site to increase data reliability. The cameras record video footage of the Chondrichthyans attracted to the bait, and additional sensors may record environmental parameters such as water temperature, depth, and visibility. The footage is reviewed to identify species based on visual characteristics and to count the maximum number of individuals of each species visible in a single frame (MaxN) to avoid recounting the same individual [14], with behaviors such as feeding, aggression, or mating displayed. Presence/absence data for each species observed is recorded, and locations where each species was observed are mapped to understand spatial distribution patterns. MaxN data is analyzed to estimate relative abundance, and statistical models can extrapolate population estimates [14]. Different behaviors observed during the recordings are categorized and quantified [15]. Species data is related to environmental variables such as habitat type, depth, and temperature to identify preferences and habitat usage, and data is compared across different times (day/night, seasons) to understand temporal patterns in occurrence and behavior [16]. BRUVS are non-invasive, allowing for ethical data collection without harming the animals or their habitat, providing several major benefits over traditional capture-based survey methods [11]: they are not size-selective (not using hooks or meshes), allow for standardized surveys (replicas can be carried out at any depth, in a variety of habitats, and by operators who do not require high levels of training) [17], and make it possible to detect even large, mobile fishes, such as many sharks and rays, that usually avoid active fishing surveys [17,18]. The use of standardized protocols for deployment and analysis enhances comparability across studies and regions, though potential biases include bait attractiveness differing by species, varying visibility conditions, and camera positioning affecting detection probabilities. BRUVS data informs species conservation status, aids in designing marine protected areas, and monitors the effectiveness of conservation measures [19]. It also helps in understanding species interactions, habitat preferences, and responses to environmental changes and in assessing the impact of fishing activities on sustainable management practices. By deploying BRUVS systematically across different regions and habitats, comprehensive data on Chondrichthyan species can be gathered, leading to a better understanding of their ecology and informing effective conservation and management strategies [20].

The use of BRUVS to study cartilaginous fish has increased significantly over the last thirty years for many of the reasons described above. Initially developed to survey marine species in hard-to-reach or sensitive environments, BRUVS offered a non-invasive alternative to traditional methods like scuba surveys and trawling. Early studies, particularly in Australia and the Pacific Islands, demonstrated BRUVS’ effectiveness in attracting and documenting a diverse array of Chondrichthyans, thereby gaining acceptance within the scientific community [18]. By the mid-2000s, BRUVS had been refined with improved camera technology and standardized protocols, facilitating broader adoption and enabling consistent, long-term data collection on Chondrichthyan populations globally [21]. The method’s versatility has proven crucial for conservation efforts, providing essential insights into species diversity, distribution, and the impacts of marine protected areas, ultimately aiding in the development of informed management strategies [20]. To date, BRUVS have been successfully used to monitor sharks, rays, and chimeras worldwide [14,20,22,23]. However, despite the increasing prevalence of this technique in the literature, a comprehensive review of the potential of BRUVS in the study of cartilaginous fish is still lacking. Given this, the aim of this study is to provide a comprehensive overview of how Chondrichthyans are studied through the BRUVS, the configurations used to target different taxa and the research objectives that can be pursued using this method. In this way, a useful tool will be made available to the scientific community, both to better understand the potential of BRUVS in the study of Chondrichthyans and to provide a quick, comprehensive guide on the different methodologies used to collect data according to the aims of the studies.

## 2. Materials and Methods

### Literature Survey and Data Selection

Searches in Google Scholar, Scopus, and ResearchGate were conducted for all articles published up to 27 October 2023, dating back to 1990. We used the keywords “baited underwater video”, “baited remote underwater video”, “baited remote underwater video system”, “baited remote underwater video station”, “BRUVS”, “BUV”, “BRUV” combined in pairs with “elasmobranchs”, “Chondrichthyans”, “sharks”, “rays”, “chimeras”. The search has returned a total of 873 findings. Only publications in peer-reviewed journals in English were selected; materials in the grey literature (i.e., technical reports, theses, etc.) were excluded from the analysis. Additionally, the references of any peer-reviewed papers considered eligible were checked in order to find further useful papers. The included studies were strictly limited to marine waters and focused on Chondrichthyans, either as the main research subjects or as incidental occurrences in other research frameworks. The reviewed studies were carried out exclusively by scientists and were field-based. Based on these selection criteria, 81 publications were gathered (Table 1).

Uniform data was recorded for every study, including year of publication, study site, type of BRUVS, additional tools, type of bait, target species/genus/family, type of information gained (qualitative, quantitative), and investigation objectives. The latter were summarized into thirteen main topics: (1) occurrence, (2) distribution, (3) abundance, (4) biomass, (5) growth, (6) size, (7) diversity, (8) density, (9) habitat suitability, (10) habitat use, (11) habitat type, (12) movements, and (13) behavior. A single study could fall into several categories of investigation. Publications lacking information on the type of BRUVS and/or three or more of the investigated parameters were excluded, as were those that used unbaited systems. This selection process may have introduced some bias into the results; however, since it led to the exclusion of fewer than 8 publications (10% of the total considered), any loss of information is deemed to be within acceptable limits. Furthermore, additional limitations may arise from the inability to capture certain publications through the search engines utilized, particularly during the first decade under investigation or earlier, as some publications may remain uncatalogued within these databases. The year 1990 was chosen as an arbitrary limit since no relevant research was produced before then. Finally, since the species *Carcharhinus limbatus* and *Carcharhinus tilstoni* were treated as a single complex (*C. limbatus*/*tilstoni*) by several authors [24,25,26,27], they have also been treated in the same way in the following results.
animals-14-01875-t001_Table 1Table 1List of peer-reviewed papers used for the present review. Legend of topics: P presence, A abundance, B biomass, DN distribution, HU habitat uses, S size, HS habitat suitability, DY diversity, DE density, TH type of habitat, M movements, G growth, BR behavior.SourceSurvey LocationType of BRUVSCamera ConfigurationsBait TypesTopics[28]South AfricabenthicStereo, mono*Sardinops sargax*P, A[29]CaribbeanbenthicStereoTunaP, A, B[30]Colombian Pacificmid-waterMonoTunaP, A[31]North Western AustraliabenthicStereo-P, DN, A[27]Australia benthicStereoPilchardsP, A, HU[32]South AfricabenthicMono*Sardinops sargax*P, A[19]North Western AustraliabenthicStereo-P, DN, A, S, HS[33]The north coast of BrittanybenthicStereo*Scomber scombrus*P, A, DY[34]BelizebenthicMonoCrushed baitfishP, DN, A[35]Western Australia drifting deep-waterStereo*Sardinops sagax*P, HS[36]South-western Pacific benthicStereo*Sardinops sagax*P, DN, DY[37]Central PacificbenthicStereoMackerelP, DN, A[11]BahamasbenthicMonoBonito tunaP, DN, A, DY, S[38]Turks and Caicos Islands benthicMono*Barracuda sphyraena*P, DN, A, DY[39]Caribbean coast of Colombia benthicMono*Opisthonema oglinum*P, A[40]Eastern Tropical Pacific, Costa Rica, Galapagos Islandsdrifting-pelagicMono*Thunnus albacares*P, DN, A[18]North-east AustraliabenthicMono*Sardinops neopilchardus*P, DN, A, DY[41]Central Mediterranean Seamid-waterMono*Sardinella aurita*P, S[42]PanamabenthicStereo*Euthynnus alletteratus*P, A, DY[43]Eastern Tropical Pacific benthicStereo*Mugil cephalus*, *Katsuwonus pelamis*P[44]Western Indian Ocean benthicMonoMackerel P, DN, A[45]Southeast Australia benthicMono-P, A, DE[26]Pacific OceanbenthicMonoClupeidae, ScombridaeP, D[46]South AfricabenthicMono*Sardinops sagax*P, DN, A, DY[47]South AfricabenthicStereo*Sardinops sagax*P, DN, A, DY[24]North-east AustraliabenthicMono*Sardinops, Sardinella* spp.P, DN, A, TH[48]Eastern Tropical Pacific benthic & mid-waterMono*Scomber japonicus*P, DN, A[49]BelizebenthicMonoSardinesP, A[50]Western Atlantic benthicMonoSardinesP[51]Fiji benthicStereo*Sardinops* sp.P, A, B[52]Southern California benthicMono*Loligo* sp.P, A[53]SeychellesbenthicMono*Scomber scombrus*, *Euthynnus affinis*P, DN, A, M[54]New South Wales, Australia benthicStereo*Sardinops neopilchardus*P, S[55]New South Wales, AustraliabenthicStereo*Sardinops neopilchardus*P, A, G, S[56]Western Australian benthicStereo*Sardinops sagax*P, A, S[57]Western Australia benthicStereo*Sardinops sagax*P, A[58]Eastern Australiamid-waterMonoMinced pilchards, bread, tuna oilP, DN, A, TH[59]Southern New Zealand downward-facing benthicMono*Thyrsites atun*, *Sardinella sirm*P, A[60]Antillesdrumline cameraMono*Sphyraena barracuda*P, BR[61]Southeast Australia benthicStereoPilchardsP, DN, DY, TH[62]Arabian Gulf benthicMono*Sardinella longiceps*P, DN, A, TH[63]South-Western Pacific benthicStereo*Sardinops sagax*P, DN, A, DY[64]New South Wales, AustraliabenthicMono*Sardinops sagax*P, A[65]French Polynesia benthicMonoSardinesP, A[66]New South Wales, AustraliabenthicStereo, Mono*Sardinops sagax*P[67]Western Atlantic benthicMono*Scomber scombrus*P, A[13]North-west AustraliabenthicStereo*Sardinops* spp.P, A, BR[68]Western Australia mid-waterStereo*Sardinops sagax*, squidP, DN, DY, S[69]New Zealand downward-facing benthicStereo-P, S[20]WorldwidebenthicMonoClupeidae, ScombridaeP, DN, A[70]Western Australia mid-waterStereoTuna, *Sardinops* sp.P, A, BR[71]Philippines benthicMono*Sphyraena barracuda*, *Caranx ignobilis*, *Serranidae* spp., *Acanthocybium solandri*, *Katsuwonus pelamis*, *Sardinus* spp.P, A, DY[23]Worldwidemid-waterStereo*Sardinops sagax*P, DN, A, S[72]South-West Australia benthicMono*Sepioteuthis australis*, *Seriola lalandi*, *Sardinops sagax*P, DN[73]Northeastern Brazil benthicMono*Sardinella brasiliensis*, *Sphyraena barracuda*P[74]South AfricabenthicMono*Sardinops sagax*P, A, DY[75]Sweden benthicStereoSardinesP, A, S[76]Central Indian Ocean drifting-pelagicMono-P[77]AntillesbenthicMono*Sarda* spp.P, BR[78]Western Mediterraneandrifting-pelagicMonoTunaP[79]North-east AtlanticbenthicMono*Scomber scombrus*P[80]BrazilbenthicStereo*Sardinella brasiliensis*, *Harengula* sp.P, A, S[81]North-west Australia benthicMono*Sardinops sagax*P, A, BR[16]Central Pacific Ocean benthicMono-P, A, BR[82]Western Australiamid-waterStereo*Mugil cephalus*P, A, DY, S[25]Mid-west coast of Western Australiamid-waterStereo*Sardinops sagax*P, A[83]Western Australiamid-waterStereo*Sardinops sagax*P[15]MassachusettsbenthicMonoMackerelP, BR[84]Malaysian BorneobenthicMono*Sardinella* spp, *Scomber australiasicus*P, A[85]IndonesiabenthicMonoClupeidae, ScombridaeP, A[14]WorldwidebenthicMono-P, DN, A, DY[86]Red Sea, SudanbenthicMono*Sarda orientalis*, *Caranx* spp., *Lutjanus* spp.P, DN, A[87]Spain, FrancebenthicMonoFlour, dried chopped fish, sardine oil, sunflower oil, sodium bicarbonate, citric acidP, A[88]Sydney metropolitan areabenthicMonoPilchards, falafel, tuna oilP, A[22]Worldwidemid-waterStereo-P, A, DY, S, B[89]Western Mediterranean Seamid-waterMonoFish scraps, cephalopods, cetacean flesh and oil P[90]WalesbenthicStereoMeal and fish oilsP, A[21]South-western Australia benthicStereo*Sardinops sagax*P, A, DE[91]North-east AustraliabenthicMono*Sardinops* spp., *Sardinella* spp.P, DN[92]New South Wales, AustraliabenthicMonoPilchard, abalone, urchinP, A[93]North-eastern New ZealandbenthicStereo*Sardinops sagax*P, DN, A, DE

## 3. Results

### 3.1. Temporal and Spatial Trends in the Use of BRUVS for Studying Cartilaginous Fish

During the last 30 years, the number of papers using BRUVS to study Chondrichthyans worldwide has grown significantly, from a single paper in the five-year period 1994–1998 to 41 papers in the period 2019–2023 (Figure 1). The origin of BRUVS for studying cartilaginous fish is Australia, where all records in the first decade of the survey came from this area, except for one record from the Northeast Atlantic [79]. However, even today, Australia remains the area where this tool has been most extensively used, followed by Central America and South Africa, with moderate use in other parts of the world (Figure 2).

Regarding the type of BRUVS, for the first two decades of the survey (up to 2014), only two configurations were used: benthic (92% of records) and mid-water (8% of records). Since 2015, new configurations have been developed to collect data on Chondrichthyans, including drifting deep-water, drifting-pelagic, and downward-facing benthic setups.

Generally, the most commonly used type of BRUVS is benthic (lying on the seabed) with 79.5% of the findings (Figure 3a), followed by mid-water or pelagic (15.7%) (Figure 3c), pelagic drifting (3.6%) and deep-water drifting (1.2%) (Figure 3b). Additionally, 1.2% of the findings involve special set-ups to meet specific study needs (e.g., drumline camera in [60]). In benthic and mid-water, BRUVS were used simultaneously to carry out field surveys [48].

The recording video configuration can be mono (one camera) (Figure 3a) or stereo (two cameras pointing at the bait bag) (Figure 3c): mono-BRUVS were used in 60.2% of the findings, only in [28,66] both configurations were used in the same field activities. Furthermore, BRUVS were used as the only survey tool in 60.1% of the reviewed papers, and in 83.1% of them, quantitative data were obtained.

### 3.2. Chondrichthyans Studied through BRUVS

The literature review revealed a census of 43 families of cartilaginous fish. The family with the highest number of records (238) is Carcharhinidae, followed by Dasyatidae (83 records) (Table 2).

A total of 195 chondrichthyan species were covered in the reviewed papers, including 99 sharks, 93 rays, and 3 chimeras. The most commonly studied species worldwide using BRUVS (with an incidence rate of more than 20%) include *Galeocerdo cuvier*, *Carcharhinus amblyrhynchos*, *Carcharhinus limbatus/tilstoni*, *Carcharhinus melanopterus*, *Sphyrna lewini*, *Triaenodon obesus* (Appendix A). BRUVS allow to collect data both on species listed Endangered (EN) and Critically Endangered (CR) (*Acroteriobatus leucospilus*, *Aetobatus narinari*, *Aetomylaeus vespertilio*, *Alopias pelagicus*, *Carcharhinus amblyrhynchos*, *C. acronotus*, *C. dussumieri*, *C. longimanus*, *C. obscurus*, *C. perezi*, *C. plumbeus*, *C. taurus*, *Centrophorus squamosus*, *Glaucostegus halavi*, *G. typus*, *Haploblepharus edwardsii*, *Himantura uarnak*, *Hypanus marianae*, *Isurus oxyrinchus*, *Leucoraja ocellata*, *Mobula birostris*, *M. kuhlii*, *M. tarapacana*, *Mustelus mustelus*, *Myliobatis aquila*, *Negaprion acutidens*, *Pristis zijsron*, *Pteromylaeus bovinus*, *Raja undulata*, *Rhina ancylostoma*, *Rhincodon typus*, *Rhinobatos hynnicephalus*, *Rhinoptera javanica*, *Rhynchobatus australiae*, *R. laevis*, *Rostroraja alba*, *Sphyrna lewini*, *S. mokarran*, *S. tiburo*, *Stegostoma tigrinum*, *Styracura schmardae*) and species considered “Data Deficient” (DD) (*Bathyraja shuntovi*, *Cirrhigaleus australis*, *Etmopterus molleri*, *Ginglymostoma unami*, *Hemitriakis abdita*, *Megatrygon microps*, *Neotrygon kuhlii*, *Taeniura lessoni*) [9].

### 3.3. Baits Used to Attract Chondrichthyans

Regarding the types of bait used to arm BRUVS to attract chondrichthyans, the most common, with 58% of records, are fish from the Clupeidae family (*Sardinops* spp., *Sardinella* spp., *Clupea* spp., *Harengula* spp.) followed by fish from the Scombridae family at 15% (*Scomber* spp., *Euthynnus* spp., *Thunnus* spp., *Katsuwonus pelamis*, *Sarda* spp., *Acanthocybium solandri*), cephalopods sensu lato 5% (e.g., *Loligo* spp., *Sepioteuthis australis*), *Sphyraena* spp. 5% (in particular *S. barracuda*), *Caranx* spp. 2% (e.g., *C. ignobilis*); the remaining 15% includes different animals such as *Opisthonema oglinum*, *Mugil cephalus*, *Thyrsites atun*, *Seriola lalandi, Serranidae* spp., abalone, urchin (Figure 4). Baits and/or their parts (viscera, oil) can be combined in different ways.

Species from the Clupeidae family have remained, over time, the preferred bait for attracting cartilaginous fish into the range of BRUVS, followed by species of the family Scombridae, which have increased their use significantly over the past decade to match the clupeids in recent years. Recently, there has also been an increase in the use of both cephalopods and species of the genus *Sphyraena* spp. Of note is the constant presence of various attractant materials, here called “other”, often associated with the above categories. The use of *Caranx* spp., detected as of 2016, remained limited (Figure 5).

### 3.4. Topics of Investigation

Using BRUVS, the three most commonly studied topics regarding Chondrichthyans are the occurrence of the species (83 papers), their abundance (61 papers) and their distribution (29 papers). Furthermore, found in more than 9 papers are diversity, size and behaviour. Marginal topics with less than 5 papers include habitat type, density, habitat suitability, biomass, movements, habitat use and growth (Figure 6).

Occurrence and abundance have been the primary topics investigated using the BRUVS, and together with distribution, they have remained the most investigated topics over time. The improvement in technology in the 21st century has allowed for a wide range of investigation topics. In recent years, this has allowed, for example, the evaluation of the size of Chondrichthyans through the use of sophisticated software, the study of their behaviors, and even the investigation of the type of habitat, its suitability, and how the species utilizes it (Figure 7).

## 4. Discussion

Baited Remote Underwater Video Systems (BRUVS) have emerged as a significant tool in marine biology for studying cartilaginous fish populations. These systems offer numerous advantages that have important implications for both scientific understanding and conservation efforts. One of their primary benefits is non-invasive monitoring, which allows for the observation of these fish in their natural environments without the stress and harm associated with traditional capture methods. This results in more accurate data collection regarding species' presence and abundance [18].

### 4.1. Implementing BRUVS

The increase in the use of BRUVS for studying marine fauna is attributed to technological advancements that have enabled the use of high-performance video recording systems with functions that go beyond simple photography or video recording: in fact, in the specific case of Chondrichthyans, it has gone from a system of a camera loaded with 800 frames of 35 mm Kodak Ektachrome color film and a reference scale in the view of the camera [79] to High-definitions stereo camera configurations, which makes it possible to obtain accurate and precise measurements of recorded specimens through specific video processing programs [13]. The significant adoption of BRUVS for studying sharks and related species has primarily occurred in Australia, South Africa, and Central America. This is certainly due to the large availability of these animals in the waters of these countries, but in the case of Australia, it is also due to the fact that it has demonstrated the potential of this tool in its waters for a long time [18,57]. In other parts of the world, such as the Mediterranean Region, where few studies have employed the use of BRUVS to study Chondrichthyans, it has only recently begun [41,78,89], the potential of this tool has yet to be recognized. BRUVS reduce labor costs and safety risks associated with deploying divers, and they provide extensive, high-resolution footage without physically disturbing marine habitats. BRUVS contributes to more comprehensive ecological assessments at a fraction of the cost and effort required by more invasive techniques [18,94].

Not all BRUVS are the same; the main differences are found in the set-up, which, in fact, depends on the objective of the research. The major differences are related to the orientation of the camera, the video recording configurations, and the position of the BRUVS in the water column. The most common camera setup is horizontal (horizontal-facing), although they may be oriented towards the seabed (downward-facing). The latter configuration can, for example, be used to allow for the photo identification of seven-gill sharks, as individuals have unique dorsal spot patterns [59]. Pioneering studies using this methodology have enabled the identification of individual white sharks and the tracking of their movements along the California coast [95], as well as the study of population dynamics and behavior of tiger sharks in Shark Bay [96]. Photo-identification is a crucial technique that enables precise calculation of population numbers and provides insights into the behavior of individual sharks and their communities.

Regarding video recording configuration, it can be mono (a single camera) or stereo (two cameras pointing at the bait bag and recording with overlapping field of view). The main difference is that stereo-BRUVS, using dual images, allow us to accurately estimate the length of fish. These lengths can be converted into biomass estimates using standardized length-weight relationships [29,51]. The most common set-up is the benthic BRUVS, lying on the bottom. This type of BRUVS allows us to record a wide range of sharks, rays [14,24,26,48], and chimeras [47,74,79,93]. Mid-water and pelagic drifting BRUVS, although less used and with fewer records of Chondrichthyans, are useful to monitor pelagic species such as *Sphyrna* spp., *Carcharhinus* spp., *Alopias pelagicus*, *Prionace glauca*, *Isurus oxyrinchus*, *Mobula birostris*, *Galeocerdo cuvier* [22,25,30,40,58,68,70,71,83]. From the standard set-ups, there are some exceptions, such as drifting deep-water BRUVS used for the construction of species distribution models [35] and baited drumline cameras used to meet special study goals, such as asses the fishing efficacy of baited hooks [60].

The type of bait used to arm BRUVS depends on the availability of what local fishermen catch; therefore, it is generally composed of Indigenous species. Regarding the family Clupeidae, the most common species used is *Sardinops sagax* [21,23,25,28,32,35,36,46,47,56,63,64,66,68,72,74,81,83,93] for the Scombridae family is *Scomber scombrus* [33,53,67,79] followed by “tunas” *Euthynnus affinis* [53], *Katsuwonus pelamis* [43,71], *Thunnus albacares* [40], *Euthynnus alletteratus* [42]. Bait can be composed either of a single species or of mixtures of species and/or their parts: in [88] the bait housing contained a 100 g mixture of minced pilchards (80 g), falafel (10 g) and tuna oil; in [89] baits used to attract sharks included fish scraps, cephalopods, and/or cetacean flesh and oil; in [73] the bait was composed of 500 g of *Sardinella brasiliensis* and barracuda viscera, *Sphyraena barracuda*.

### 4.2. Informing Conservation Initiatives

BRUVS play a crucial role in gathering essential data on species critical to conservation efforts, including those classified as Critically Endangered, Endangered, and Data Deficient. By providing a non-invasive method to observe these species in their natural environments, BRUVS allow for the detection and monitoring of populations that are otherwise difficult to study due to their rarity or elusive behavior. This technology facilitates the gathering of essential information on species distribution, abundance, and ecological interactions, which is vital for assessing conservation status and implementing protective measures. For instance, studies have demonstrated the effectiveness of BRUVS in documenting the presence of rare and threatened species, thereby filling significant knowledge gaps and aiding in the development of targeted conservation strategies [18,97].

BRUVS are instrumental in documenting species diversity and distribution across various habitats and depths, which is essential for mapping the geographical range of species and understanding their ecological niches. This information can inform habitat protection strategies [94]. From the video recordings made with BRUVS, the most commonly extrapolated data is the species’ relative abundance. To calculate this, it is commonly used MaxN, considered the maximum number of each species observed in a single frame of each 60-min deployment, then averaged across all deployments on one site [14]. The distribution of a species can be assessed by carrying out several BRUVS sessions at different sites within a given area [20]. The study of the diversity of chondrichthyans is closely related to estimates of their occurrence, although the association of BRUVS with other monitoring techniques has shown more accurate estimates for this parameter [36]. More accurate size estimation is related to the use of stereo-BRUVS; in fact, by processing videos from two different points of view through dedicated software (i.e., SeaGIS EventMeasure software http://www.seagis.com.au), it is possible to estimate different morphometric parameters such as snout-to-first dorsal fin length, dorsal fin-to- precaudal length, fork length (FL), precaudal length (PCL) and total length (TL) [54]. Furthermore, BRUVS provide detailed insights into fish behavior, including inter and intraspecific interactions [16], behavioral change [70], foraging behaviors [77].

The study of habitat preferences is another area where BRUVS excel, revealing the importance of different environmental features for various species. This knowledge is crucial for understanding habitat use and the ecological requirements of cartilaginous fish [58]. In terms of conservation, data from BRUVS can inform the design and management of Marine Protected Areas (MPAs) by identifying critical habitats and biodiversity hotspots, thereby enabling more effective targeted conservation efforts [98]. Additionally, BRUVS enable long-term monitoring programs, essential for tracking the effectiveness of conservation measures over time. Consistent data collection allows for the assessment of trends and the adaptation of strategies to ensure the ongoing protection of cartilaginous fish populations [99].

### 4.3. Challenges and Considerations

BRUVS can be used as a sole survey tool, as they offer a non-invasive alternative to other survey instruments to monitor trends in the relative abundance of Chondrichthyans [11], or associated with other traditional tools (longline, beach seine, visual census, interviews, audible-stationary-count, Environmental DNA) to provide a more comprehensive and exhaustive assessment of life traits aspects in different marine conditions [18,39,44,45,52,81,86]. Innovative methods such as autonomous marine vehicles (i.e., biomimetic robotics [100]), used for approaching and investigating fish in natural and aquafarming environments, could be paired with BRUVS, as they are becoming essential tools in aquatic environmental monitoring systems [101].

Despite their benefits, the implementation of BRUVS presents challenges. The initial equipment cost and the need for technical expertise can be barriers for some research programs. Moreover, interpreting video data can be time-consuming and requires careful analysis to ensure accuracy. One major advantage of using bait in BRUVS is the increased likelihood of attracting a diverse array of species, including elusive or nocturnal ones, thereby enhancing the representativeness of collected data. However, the type and quantity of bait used can be contentious. Overuse or inappropriate selection of baits can result in biased results, attracting some species disproportionately while others remain under-represented [102]. Environmental factors such as current and topography also play a critical role in bait effectiveness. Strong currents can disperse bait scent too quickly, reducing its attractant potential, while complex topography might create bait traps, where the bait scent is confined to a specific area, influencing fish behavior and the likelihood of detection [58]. These factors must be carefully managed to ensure the reliability and accuracy of data obtained through BRUVS.

## 5. Conclusions

The utilization of baited remote underwater video systems (BRUVS) represents a valuable and increasingly popular method for studying Chondrichthyans in their natural environments. Representing a non-invasive sampling method, BRUVS offer a unique opportunity to investigate up to thirteen different aspects of the life histories of these elusive marine species, providing insights that are difficult to obtain through traditional, often invasive, sampling methods. Furthermore, BRUVS allow to collect data over large spatial and temporal scales, facilitating comprehensive assessments of sharks, rays and chimeras populations and their ecological roles. This review highlights significant trends and developments over the past 30 years. Initially, BRUV studies were rare, with only one paper between 1994–1998, but their use has grown exponentially, reaching 41 papers from 2019–2023. Australia was the pioneer in using BRUVS, and it remains the most active region, followed by Central America and South Africa, with moderate use in other regions. Since 2015, BRUVS configurations have evolved from primarily benthic and mid-water setups to include drifting deep-water, drifting-pelagic, and downward-facing benthic configurations. Benthic BRUVS remain the most commonly used, followed by mid-water, pelagic drifting, and deep-water drifting. Video recording setups are mainly mono, with stereo configurations used in a minority of cases. The primary research topics investigated using BRUVS are the occurrence, abundance, and distribution of Chondrichthyans. Technological advancements have expanded the scope of research, allowing for detailed studies on size, behavior, habitat type, and habitat suitability. BRUVS have been used to study 43 families and 195 species of cartilaginous fish. The most frequently studied families are Carcharhinidae and Dasyatidae. BRUVS play a crucial role in collecting data on endangered and critically endangered species, and those considered data deficient. In terms of bait, Clupeidae species are the most common, followed by Scombridae, cephalopods, *Sphyraena* spp., and *Caranx* spp. The use of clupeids has remained consistent, with a notable increase in scombrids in recent years.

The significance of this study lies in highlighting the growing and evolving use of BRUVS as a non-invasive tool for Chondrichthyan research. This method provides critical data for conservation efforts, helping to monitor species occurrence, abundance, and distribution and contributing to the understanding of habitat use and suitability. The advancements in BRUVS technology and methodology enhance the ability to study and protect these vital marine species. Future research and methodological improvements to study Chondrichthyans could focus on several key areas. First, optimizing bait type and quantity is crucial, but exploring alternative baits that may attract different species or age classes of cartilaginous fish could provide more comprehensive data [103]. Second, enhancing camera technology and deployment strategies can improve data quality. Incorporating higher-resolution cameras and more robust, 360-degree recording capabilities could better capture rapid or elusive movements typical of these species [91]. Third, conducting studies across various habitats and depths can address knowledge gaps about cartilaginous fish distribution and behavior in less-explored regions [104]. Finally, standardizing methodologies across studies would facilitate better comparisons and meta-analyses, leading to a broader understanding of cartilaginous fish ecology and conservation needs [99]. Despite some limitations, such as potential biases in bait attraction, the utility of BRUVS in this field of research cannot be overstated. As technology and methodologies continue to improve, BRUVS are expected to play an increasingly prominent role in furthering our understanding of Chondrichthyes and sustaining conservation actions aimed at protecting these crucial components of marine ecosystems.

## Figures and Tables

**Figure 1 animals-14-01875-f001:**
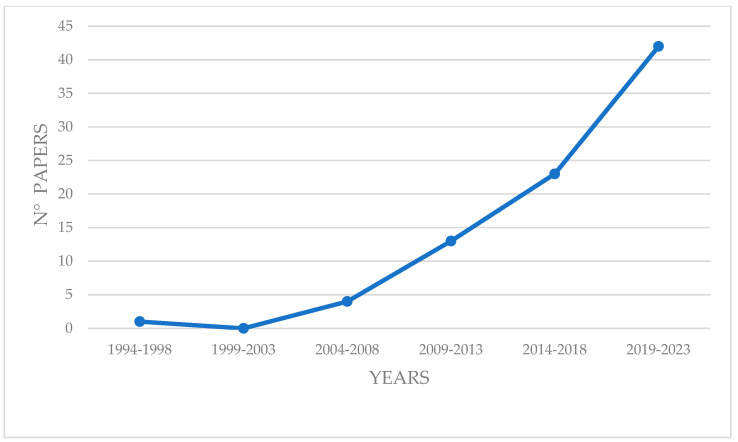
Temporal distribution of the last 30 years of papers that used BRUVS to study Chondrichthyans worldwide.

**Figure 2 animals-14-01875-f002:**
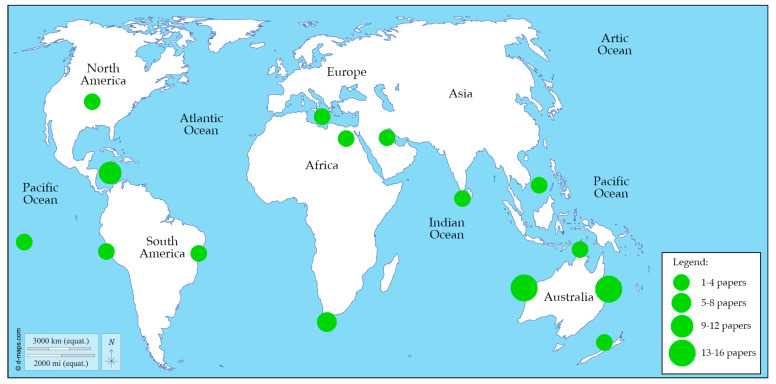
Geographical distribution of the last 30 years of papers that used BRUVS to study Chondrichthyans worldwide.

**Figure 3 animals-14-01875-f003:**
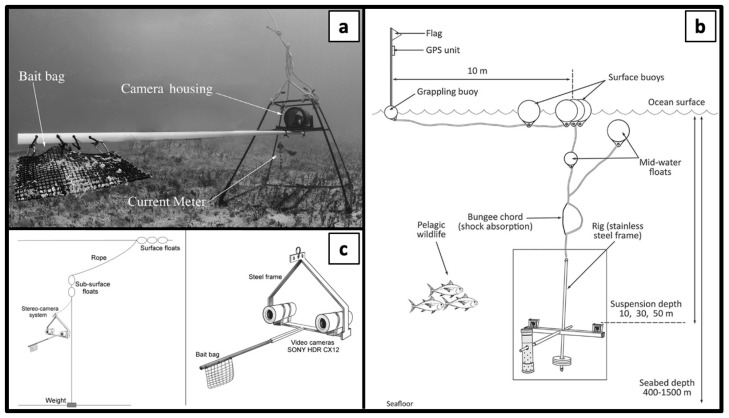
Type of BRUVS: (**a**) benthic [11]; (**b**) deep-water drifting [35]; (**c**) mid-water or pelagic [83].

**Figure 4 animals-14-01875-f004:**
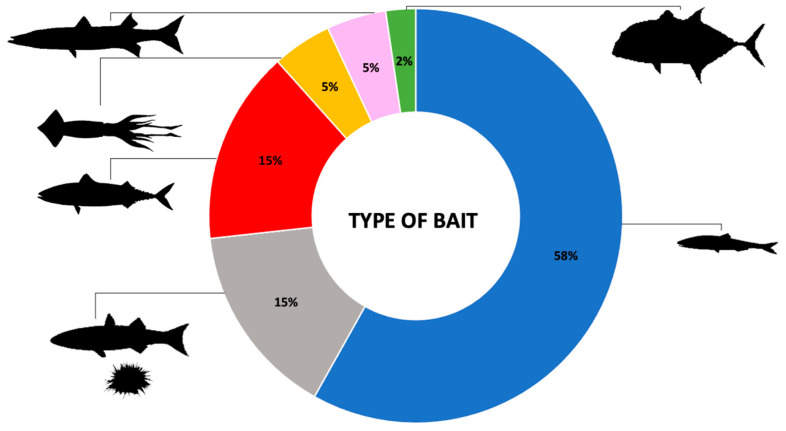
Type of bait used for arming BRUVS.

**Figure 5 animals-14-01875-f005:**
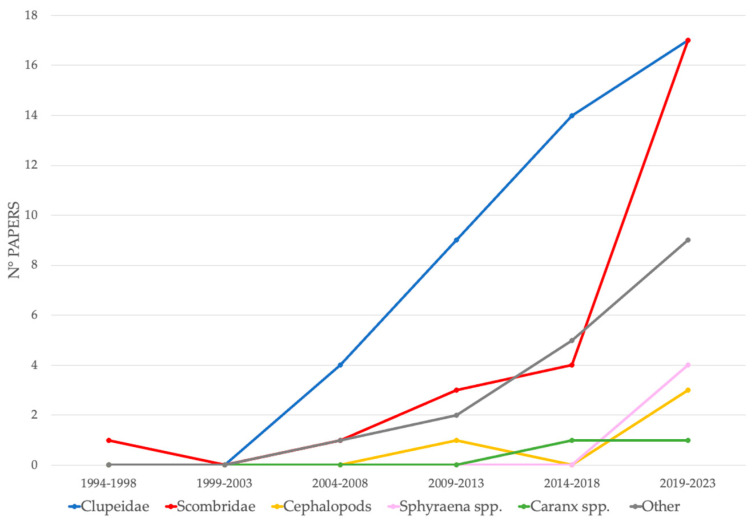
Temporal trend of the type of bait used for arming BRUVS. Bait categories are shown with different colors.

**Figure 6 animals-14-01875-f006:**
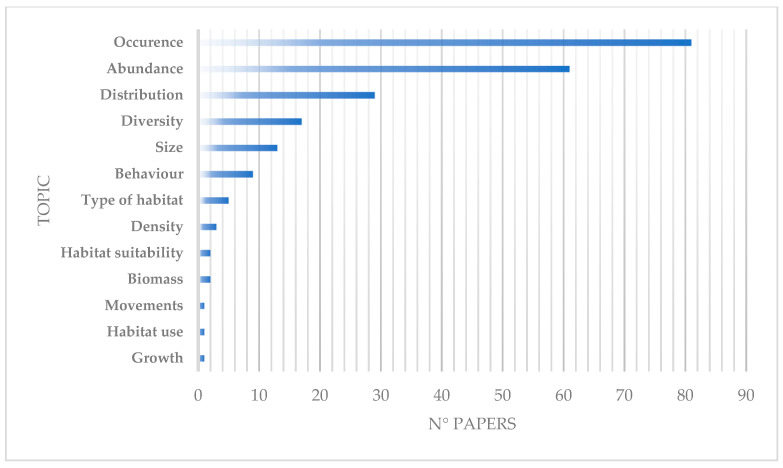
Topics studied in Chondrichthyans using BRUVS.

**Figure 7 animals-14-01875-f007:**
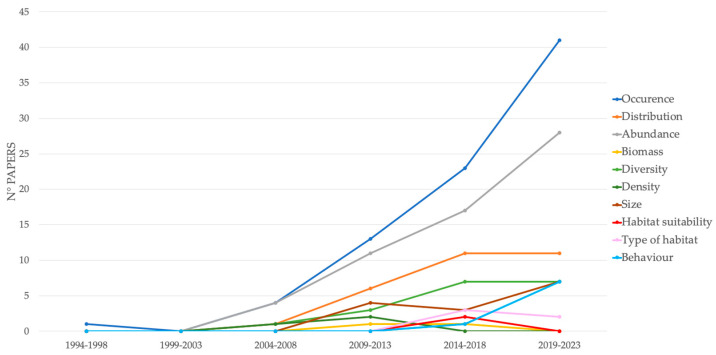
The temporal trend of the topics investigated using BRUVS (Movements (2020), Habitat use (2022) and Growth (2019) were not included in the graph since they had only 1 record).

**Table 2 animals-14-01875-t002:** Families surveyed with BRUVS: the numbers of records recorded in the literature reviewed are given.

Family	n° Records
Carcharhinidae	238
Dasyatidae	83
Sphyrnidae	37
Scyliorhinidae	32
Triakidae	32
Myliobatidae	28
Ginglymostomatidae	23
Rhinobatidae	19
Rajidae	13
Lamnidae	12
Rhinidae	11
Squalidae	11
Hemigaleidae	8
Hexanchidae	8
Stegostomatidae	8
Hemiscylliidae	7
Mobulidae	6
Urotrygonidae	6
Alopiidae	5
Etmopteridae	5
Heterodontidae	5
Orectolobidae	5
Rhincodontidae	5
Aetobatidae	4
Glaucostegidae	4
Urolophidae	4
Chimaeridae	3
Gymnuridae	3
Narcinidae	3
Odontaspididae	3
Potamotrygonidae	3
Somniosidae	3
Callorhinchidae	2
Arhynchobatidae	1
Centrophoridae	1
Dalatiidae	1
Platyrhinidae	1
Pristidae	1
Pristiophoridae	1
Pseudotriakidae	1
Rhinochimaeridae	1
Squatinidae	1
Trygonorrhinidae	1

## Data Availability

No new data were created.

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
