# Peer review of "Studying Chondrichthyans Using Baited Remote Underwater Video Systems: A Review"

_animals, 2024, doi:10.3390/ani14131875_

Round 1
Reviewer 1 Report
Comments and Suggestions for Authors
The manuscript reviews of types of underwater camera trap use, which could be a useful reference for new users to the system. As the authors point out, it is an established technique for at least 34 years now.
The manuscript could benefit from a discussion, even brief, of the benefits and impairments of using bait. This can be contentious with the amount and type of bait, current and topography affecting the effectiveness of bait.
There are some tables that could be removed without diminishing the manuscript’s focus. Some specific points below will hopefully be useful:
L98: Table 1 could be summarised or included as supplementary material.
It would also benefit by including photoidentification as a category, which is an important use of BRUVS
Figure 2 & 4: very interesting figures
L147 what does “the incidence of the total” mean?
Table 2 is not very useful, but perhaps might be included as supplementary material.
L170: it would be useful to mention the cost and non-invasiveness of BRUVS compared to other methods.
L228: what does strategic mean here?
Conclusions: bait is first discussed here and this needs to be introduced at the beginning of the manuscript.
L265: any Acknowledgements?
Comments on the Quality of English LanguageThe grammar and punctuation need some revision to make the manuscript clearer, a few examples are:
L48: what does assed mean? Used further on in the manuscript as well.
L 60: the sentence is missing a conjunction to make sense?
L78: remove the
L 154 is whit with? Used further on in the manuscript as well.
Discussion & Conclusions: the grammar needs to be revised throughout these sections to make the manuscript clear.
Author Response
Request: The manuscript reviews of types of underwater camera trap use, which could be a useful reference for new users to the system. As the authors point out, it is an established technique for at least 34 years now. The manuscript could benefit from a discussion, even brief, of the benefits and impairments of using bait. This can be contentious with the amount and type of bait, current and topography affecting the effectiveness of bait.
Answer: We are gratefull to reviewer 1 for his/her comments. We made suggested changes as it is evident in the discussion section.
Request: There are some tables that could be removed without diminishing the manuscript’s focus.
Answer: we removed table 2 from the paper and we proposed this table as a supplementary matherial
Some specific points below will hopefully be useful:
Request L98: Table 1 could be summarised or included as supplementary material. It would also benefit by including photoidentification as a category, which is an important use of BRUVS
Answer: Since other 2 reviewers asked us to improve and maintain this table, we decided to maintain it in the paper. We discussed about photoidentification in line 451
Request: L147 what does “the incidence of the total” mean?
Answer: As it may cause confusion, we have removed the term
Request: Table 2 is not very useful, but perhaps might be included as supplementary material.
Answer: Done
Request: L170: it would be useful to mention the cost and non-invasiveness of BRUVS compared to other methods.
Answer: done, see lines 442-445
Request: L228: what does strategic mean here?
Answer: As it may cause confusion, we have removed the term
Request: Conclusions: bait is first discussed here and this needs to be introduced at the beginning of the manuscript.
Answer: Many thanks for this suggestion, we added a part in the introduction (line: 90) and discussion (lines 334-359)
Request: L265: any Acknowledgements?
Answer: we do not have any
Request: L48: what does assed mean? Used further on in the manuscript as well.
Answer: done! Changed as assessed
Request: L 60: the sentence is missing a conjunction to make sense?
Answer: done, we changed all the paragraph
Request: L78: remove the
Answer: done, we changed all the paragraph
Request: L 154 is whit with? Used further on in the manuscript as well.
Answer: done
Request: Discussion & Conclusions: the grammar needs to be revised throughout these sections to make the manuscript clear.
Answer: done
Reviewer 2 Report
Comments and Suggestions for Authors
The paper “Studying Chondrichthyans using Baited Remote Underwater 2 Video Systems: a review” by Leonetti et al. examines the use of Baited Remote Underwater Video Systems (BRUVS) as a non-invasive method for studying cartilaginous fish populations.
The work needs major revisions to be considered for acceptance.
The introduction provides an overview of the current state of cartilaginous fish populations worldwide, emphasizing the decline due to overfishing and slow reproductive rates.
Clarify the significance of the study in addressing the lack of basic ecological information on cartilaginous fish and its implications for conservation efforts.
Concernign the literature review methodology, provide details on the search engines used, keywords employed, and criteria for selecting relevant papers.
Also, mention any limitations or biases in the literature review process to enhance the transparency of the study.
Expand on the description of BRUVs (baited remote underwater video systems) to ensure readers understand its non-invasive nature and advantages over traditional survey methods.
Include a brief historical background on the use of BRUVs in studying cartilaginous fish to provide context for the study.
Provide detailed information on the setup and deployment of BRUVs, including camera configurations, bait types, and survey locations.
Clarify how data on species occurrence, distribution, abundance, and behavior were collected and analyzed using BRUVs.
Present findings from the literature review in a clear and organized manner, highlighting key trends in the use of BRUVs for studying cartilaginous fish.
Also, authorsh should include innovative methods, beside BRUV or where BRUV may be integrated, such as biomimetic robotics used to approach and investigate fish in natural and aquafartming environments. Some relevant studies
Manduca, G., Padovani, L., Carosio, E., Graziani, G., Stefanini, C., & Romano, D. (2023, November). Development of an Autonomous Fish-Inspired Robotic Platform for Aquaculture Inspection and Management. In 2023 IEEE International Workshop on Metrology for Agriculture and Forestry (MetroAgriFor) (pp. 188-193). IEEE.
Bayat, B., Crespi, A., & Ijspeert, A. (2016, November). Envirobot: A bio-inspired environmental monitoring platform. In 2016 Ieee/Oes Autonomous Underwater Vehicles (Auv) (pp. 381-386). IEEE.
Interpret the findings in the context of existing knowledge on cartilaginous fish ecology and conservation.
Discuss the implications of using BRUVs for advancing our understanding of cartilaginous fish populations and informing conservation initiatives.
Address any limitations or challenges associated with the use of BRUVs and suggest areas for future research or methodological improvements.
Summarize the main findings of the study and reiterate its significance for cartilaginous fish conservation. Provide recommendations for future research directions or practical applications of BRUVs in monitoring and managing cartilaginous fish populations.
A deep English revision is needed
Comments on the Quality of English LanguageSee general comments.
Author Response
Request: The paper “Studying Chondrichthyans using Baited Remote Underwater 2 Video Systems: a review” by Leonetti et al. examines the use of Baited Remote Underwater Video Systems (BRUVS) as a non-invasive method for studying cartilaginous fish populations. The work needs major revisions to be considered for acceptance.
Answer: we thank the reviewer for the suggestions and the time spent for the revision
Request: The introduction provides an overview of the current state of cartilaginous fish populations worldwide, emphasizing the decline due to overfishing and slow reproductive rates. Clarify the significance of the study in addressing the lack of basic ecological information on cartilaginous fish and its implications for conservation efforts.
Answer: done, see lines 40-51 and lines 62-98 and lines 101-111
Request: Concernign the literature review methodology, provide details on the search engines used, keywords employed, and criteria for selecting relevant papers. Also, mention any limitations or biases in the literature review process to enhance the transparency of the study.
Answer: done, see lines 129-210
Request: Expand on the description of BRUVs (baited remote underwater video systems) to ensure readers understand its non-invasive nature and advantages over traditional survey methods.
Answer: done, see lines 62-98 and lines 101-111
Request: Include a brief historical background on the use of BRUVs in studying cartilaginous fish to provide context for the study.
Answer: done, see lines 101-108
Request: Provide detailed information on the setup and deployment of BRUVs, including camera configurations, bait types, and survey locations.
Answer: done, see table 1
Request: Clarify how data on species occurrence, distribution, abundance, and behavior were collected and analyzed using BRUVs.
Answer: done, see the Discussion section
Request: Present findings from the literature review in a clear and organized manner, highlighting key trends in the use of BRUVs for studying cartilaginous fish.
Answer: done, see the Result section
Request: Also, authorsh should include innovative methods, beside BRUV or where BRUV may be integrated, such as biomimetic robotics used to approach and investigate fish in natural and aquafartming environments. Some relevant studies
Manduca, G., Padovani, L., Carosio, E., Graziani, G., Stefanini, C., & Romano, D. (2023, November). Development of an Autonomous Fish-Inspired Robotic Platform for Aquaculture Inspection and Management. In 2023 IEEE International Workshop on Metrology for Agriculture and Forestry (MetroAgriFor) (pp. 188-193). IEEE.
Bayat, B., Crespi, A., & Ijspeert, A. (2016, November). Envirobot: A bio-inspired environmental monitoring platform. In 2016 Ieee/Oes Autonomous Underwater Vehicles (Auv) (pp. 381-386). IEEE.
Answer: done
Request: Interpret the findings in the context of existing knowledge on cartilaginous fish ecology and conservation.
Answer: done, see the Discussion section
Request: Discuss the implications of using BRUVs for advancing our understanding of cartilaginous fish populations and informing conservation initiatives.
Answer: done, see lines 548-590
Request: Address any limitations or challenges associated with the use of BRUVs and suggest areas for future research or methodological improvements.
Answer: done, see lines 592-617
Request: Summarize the main findings of the study and reiterate its significance for cartilaginous fish conservation. Provide recommendations for future research directions or practical applications of BRUVs in monitoring and managing cartilaginous fish populations.
Answer: done, see lines 629-647
Request: A deep English revision is needed
Answer: done
Reviewer 3 Report
Comments and Suggestions for Authors
General appraisal:
Interesting and useful manuscript deserving to be published.
Specific comments:
Keywords: They largely repeat the title instead of complementing it as they should.
Lines 65-66: But is can also be problematic when trying to make abundance estimations as different species have differencial predatory behaviours and be differentially attracted to BRUVS. So, a cautionary note about this constraint should be introduced here.
Lines 150-151: Give also here the numbers for these three topics.
Author Response
Request: General appraisal: Interesting and useful manuscript deserving to be published.
Answer: we thank the reviewer for his/her appreciation
Request: Keywords: They largely repeat the title instead of complementing it as they should.
Answer: done
Request: Lines 65-66: But is can also be problematic when trying to make abundance estimations as different species have differencial predatory behaviours and be differentially attracted to BRUVS. So, a cautionary note about this constraint should be introduced here.
Answer: done
Request: Lines 150-151: Give also here the numbers for these three topics.
Answer: done
Round 2
Reviewer 1 Report
Comments and Suggestions for Authors
The authors have made substantial changes to the manuscript which have enhanced the paper. The emphasis on Australian work seemed to limit the authors somewhat, but they have now included a small amount on photoidentification and on the use of bait to broaden the scope of the review. Table 1 is a great improvement. The paper now needs to be edited for grammar to allow readers to understand what the authors are trying to relay. Some specific areas include:
Grammar incorrect in Line 61 histories is not Genitive, Line 82, 84, 93, 109, 136, 37, 138, 139, 141, 148, 155 (twice), 156, 193, 200, 207, 210, 237, 257, 258 (awkward wording), 293, 399 (repeated from Line 335), 344, sentence 378, 389.
Line 118: the goal of the paper written here is not what the title or the rest of the paper is about.
References are missing from Line 186, 378
Table 2, Fig. 5: need more information in the caption – captions are meant to allow the reader to understand what the table or figure is trying to convey without resorting to the body of the text. Definitions are needed to explain what the terms included mean.
Fig. 7 is interesting but not really necessary for the review.
Line 299 As this is a review paper and photoidentification is a key topic for use in BRUVS, the authors should acknowledge the pioneers in this area rather than mention a single very recent paper from the early 2000s. It is an important technique that allows accurate population numbers to be calculated as well as an understanding of individual sharks and community behaviour.
Line 348 Reference 96 is about acoustic tagging, not the use of video material. The sentence is out of place or needs to be rewritten.
Comments on the Quality of English LanguagePlease see my comments above.
Author Response
Comments 1: The authors have made substantial changes to the manuscript which have enhanced the paper. The emphasis on Australian work seemed to limit the authors somewhat, but they have now included a small amount on photoidentification and on the use of bait to broaden the scope of the review. Table 1 is a great improvement. The paper now needs to be edited for grammar to allow readers to understand what the authors are trying to relay.
Response 1: First of all, many thanks to the reviewer for his suggestions! We have edited the paper asking to a native-English speaking reader.
Comments 2: Grammar incorrect in Line 61 histories is not Genitive, Line 82, 84, 93, 109, 136, 37, 138, 139, 141, 148, 155 (twice), 156, 193, 200, 207, 210, 237, 257, 258 (awkward wording), 293, 399 (repeated from Line 335), 344, sentence 378, 389.
Response 2: Done!
Comments 3: Line 118: the goal of the paper written here is not what the title or the rest of the paper is about.
Response 3: We deleted the phrase.
Comments 4: References are missing from Line 186, 378
Response 4: Done!
Comments 5: Table 2, Fig. 5: need more information in the caption – captions are meant to allow the reader to understand what the table or figure is trying to convey without resorting to the body of the text. Definitions are needed to explain what the terms included mean.
Response 5: We added definitions and improved the caption.
Comments 6: Fig. 7 is interesting but not really necessary for the review.
Response 6: Since the other 2 reviewers asked for this figure, we preferred to keep it in the paper.
Comments 7: Line 299 As this is a review paper and photoidentification is a key topic for use in BRUVS, the authors should acknowledge the pioneers in this area rather than mention a single very recent paper from the early 2000s. It is an important technique that allows accurate population numbers to be calculated as well as an understanding of individual sharks and community behaviour.
Response 7: We improved the text: see lines 741-746.
Comments 8: Line 348 Reference 96 is about acoustic tagging, not the use of video material. The sentence is out of place or needs to be rewritten.
Response 8: We deleted the phrase.